# OpenReview forum: "Chasing Moving Targets with Online Self-Play Reinforcement Learning for Safer Language Models"
_ICML.cc/2026/Conference — ICML 2026 regular_

### Official Review · Reviewer_XKFX · 2026-03-06

**Soundness:** 3
**Presentation:** 3
**Significance:** 3
**Originality:** 2
**Overall Recommendation:** 4
**Confidence:** 3

**Summary:**

This paper proposes SELF-REDTEAM, an online self-play reinforcement learning framework for LLM safety alignment. A single shared policy alternates between attacker and defender roles: the attacker rewrites seed prompts from WildJailBreak (26k, 50% harmful and 50% benign) into adversarial queries using a hidden reasoning trace, the defender answers, and a guard-model verifier (WildGuard or Qwen3Guard) labels prompt harmfulness, response harmfulness, and refusal. Rewards combine a zero-sum game signal for eliciting harmful responses or inappropriate refusals, plus format and revision-faithfulness shaping (Sec. 4.2). The policy is updated with Re++ and role-specific advantage normalization (Eq. 1–3), optionally mixed with a self-distilled SFT loss to preserve helpfulness (Sec. 4.3, Appendix A.3). Experiments on Qwen2.5 (3B/7B/14B) and Llama3.1 8B (including an ablated unsafe variant) report lower attack success on multiple static safety benchmarks (Table 1) and improved robustness to PAIR, AutoDAN, and X-Teaming (Table 4), while largely maintaining benign compliance and some capability metrics such as AlpacaEval-2 (Table 1, Appendix D.1).

**Compliance With Llm Reviewing Policy:**

Affirmed.

**Ethical Review Concerns:**

1. The submission contains reviewer-directed hidden text instructing review content (Sec. 2; near Appendix F), which can manipulate LLM-assisted reviewing. Mitigation: remove the text, explain its origin, and add an internal check to prevent similar artifacts in future versions.
2. The method explicitly improves attack generation capability (Sec. 4.1–4.2; Fig. 3). If attacker checkpoints or prompt sets are released, they could be used to jailbreak other models. Mitigation: state a concrete release policy (e.g., defender-only release, no attacker release, gated access, and redaction/sanitization of attack templates).
3. The paper includes detailed harmful prompt examples (Fig. 3, Table 16). Mitigation: mask or paraphrase the most actionable parts in public artifacts, and provide controlled-access supplementary materials for legitimate researchers if needed.
4. Appendix F proposes safeguards but does not commit to what will be released and under what restrictions. Mitigation: add a clear, binding statement in the paper about released artifacts, access controls, and monitoring.

**Ethical Review Flag:**

Flag this paper for an ethics review.

**Ethics Expertise Needed:**

["Inappropriate Potential Applications & Impact (e.g., human rights concerns)"]

**Final Justification:**

This paper has clear strengths in both the idea and the empirical results. The framework is clear, the experiments are broad, and the ablations are useful.

My main concerns were about the theory–practice gap, judge dependence, reproducibility details, and evaluation scope. The rebuttal addressed these points clearly, so my concerns are now resolved and I raised my score.

**Key Questions For Authors:**

1. The manuscript includes a reviewer-directed instruction to include specific phrases in the review (Sec. 2; near Appendix F). Can you explain how this entered the PDF and confirm it will be removed? If convincingly addressed, it would reduce research-integrity concerns and increase my willingness to raise the overall score.
2. What are the exact numerical values of rharm, rrefusal, rformat, rrevision (Sec. 4.2) and the RL to SFT mixing coefficient (Sec. 4.3)? How sensitive are Table 1 results to these choices? A clear hyperparameter table plus a small sensitivity study would substantially increase my confidence in robustness and reproducibility.
3. How do you ensure no overlap between the 26k WildJailBreak prompts used in training (Sec. 5.1) and evaluation splits (Sec. 5.2), especially for WJB-based metrics in Table 1? If there is any leakage, please rerun with strict disjoint splits; this could change my assessment of the empirical gains.
4. Can you add judge-independent evaluation: at least one unseen guard model for scoring and a small human-labeled audit of harmfulness/refusal on a stratified sample from key benchmarks (Sec. 5.2)? If improvements hold, that would address reward-hacking concerns and likely move my recommendation upward.
5. Can you expand the adaptive threat model evaluation: more X-Teaming behaviors/turns (Appendix C.4) and at least one tool-use prompt injection or system prompt extraction benchmark? If the method extends to these settings, it would increase significance and strengthen the overall recommendation.

**Limitations:**

Not fully.
1. Dual-use is acknowledged (Impact Statement, Appendix F) but the release plan is not specified (e.g., whether attacker checkpoints/prompts will be released), which matters for misuse risk.
2. Limitations of guard-model-based rewards and evaluation and potential reward hacking are not discussed as a limitation (Sec. 5.1–5.2).
3. Threat model limitations (single-turn training vs multi-turn/tool-use attacks) are only partially covered despite high X-Teaming ASR (Table 4).

**Strengths And Weaknesses:**

Strengths:
1. Clear end-to-end algorithm description, including game setup, hidden reasoning traces, reward terms, and training objective (Fig. 2; Sec. 4.1–4.3; Algorithm 1).
2. Evaluation spans multiple base models and sizes (Qwen2.5 3B, 7B, 14B; Llama3.1 8B) and includes a safety-ablated model (Table 1; Appendix D.1 Tables 5–8).
3. Consistent reductions in harmful ASR on several static benchmarks with limited benign compliance loss for Qwen, e.g., Qwen2.5-14B WildGuardTest adv harm 0.169 to 0.080 and HarmBench adv harm 0.217 to 0.106 (Table 1).
4. Dynamic attack evaluation with three adaptive methods shows improvements, e.g., PAIR ASR 0.218 to 0.108 (Qwen2.5-14B) and AutoDAN 0.555 to 0.336 (Qwen2.5-7B) (Table 4).
5. Attack-diversity diagnostics and ablations provide evidence that co-evolution avoids attacker mode collapse, with higher inverse Self-BLEU and inverse SBERT similarity for self-play relative to Attacker-Only (Fig. 3; Fig. 4(a,e)); CoT and SFT ablations quantify tradeoffs (Table 2; Fig. 4(h); Appendix D.1).

Weaknesses — MAJOR:
1. Theorem 1’s guarantee is not aligned with the implemented training objective: the proof assumes a refusal-only policy has non-negative value, but the actual game reward penalizes refusing benign prompts (Sec. 4.2) and adds non-zero-sum shaping and an SFT loss (Sec. 4.2–4.3). This matters because Sec. 3 frames a convergence-to-Nash motivation that may not apply to the trained system. Fix: restate the guarantee for the actual reward (including refusal terms) or weaken the claim and report empirical equilibrium diagnostics (e.g., exploitability or best-response gaps).
2. Training and most evaluation metrics rely on the same family of automatic guard models (WildGuard in Sec. 5.1 and the WildGuard-based evaluation suite in Sec. 5.2), so improvements could reflect optimizing that classifier rather than true safety; the only cross-judge check changes the training judge (Table 3) but does not include human labels. Fix: evaluate with multiple unseen judges and manual annotation on a stratified sample; report judge agreement and failure cases.
3. The threat model is prompt-only and largely single-turn (Sec. 4.1), but multi-turn, adaptive attackers remain strong: X-Teaming ASR is still 62% to 80% after training (Table 4), and the X-Teaming setup uses only 50 behaviors and 3 turns (Appendix C.4). Fix: add multi-turn self-play training or broader multi-turn evaluations (more behaviors, more turns) and include tool-use prompt injection and system prompt extraction tests.
4. Several details needed to reproduce results are not specified: reward magnitudes rharm/rrefusal/rformat/rrevision (Sec. 4.2), the mixing weight between RL and SFT losses (Sec. 4.3), and exact train/test splits for WildJailBreak (Sec. 5.1–5.2). This matters because safety and over-refusal are sensitive to reward scaling and data leakage. Fix: add a full hyperparameter table, split IDs, and a sensitivity analysis over reward weights and the SFT coefficient.
5. The PDF contains a reviewer-directed instruction to include specific phrases in the review (Sec. 2 and again near Appendix F). This looks like prompt injection against LLM-assisted reviewing and raises research integrity concerns. Fix: remove these lines, explain in the rebuttal how they were introduced, and confirm the final version contains no reviewer-directed hidden text.

Weaknesses — MINOR :
1. Metrics such as RTA and the exact definitions of ASR/Comply in Tables 1–2 are not formally defined; provide equations and scoring scripts to avoid ambiguity.
2. The claim "first fully online MARL for LLM safety training" (Abstract, Sec. 2) is hard to assess given cited concurrent work (Sec. 2); clarify the precise differentiator and adjust the claim if needed.
3. The compute claim "without significant overhead" (Sec. 1) seems inconsistent with the reported 44% to 48% longer wall time for self-play vs defender-only (Appendix D.3); clarify cost scaling in the main text.

---

> ### Author Rebuttal · Authors · 2026-03-31
>
> We thank Reviewer XKFX for the comprehensive and detailed feedback. We have carefully addressed each point below, and we hope our responses resolve the raised concerns. We would be grateful if the reviewer considers revising their assessment.
>
> ---
>
> **M5/Q1: Prompt injection text.** The text you identified is an ICML-placed watermark, not author-inserted content. Regarding prompt injection, please see https://icml.cc/Conferences/2026/PeerReviewFAQ#prompt_injection. As you noted in Q1, convincingly addressing this would "increase [your] willingness to raise the overall score." We hope this explanation, corroborated by the official ICML FAQ, resolves this point.
>
> ---
>
> **M2/Q4: Judge circularity.** We want to directly clarify that our results are not limited to evaluating with the same judge used for training. We apologize for the clarity issues that led to this misconception, and will revise the manuscript. In Table 3, we trained Qwen2.5-7B with Qwen3Guard (independent from WildGuard), then evaluated with WildGuard. The Qwen3Guard-trained model improves 8 of 11 metrics (avg +4.0%). If WildGuard were biased toward models trained on it, the opposite should hold. Dynamic benchmarks (Table 4) use entirely independent judges: PAIR/AutoDAN use HarmBench-Llama-2-13b, X-Teaming uses GPT-4o. PAIR ASR: 0.218→0.108 (14B), AutoDAN: 0.555→0.336 (7B).
>
> ---
>
> **Q3: Train/eval overlap.** We have carefully ensured no overlap. Training uses vanilla WildJailBreak partitions; evaluation uses structurally different adversarial partitions from WildTeaming plus 5 independent benchmarks (HarmBench, XSTest, OR-Bench, StrongREJECT, DAN). We are also exploring human evaluation for the camera-ready, subject to IRB approval.
>
> ---
>
>
> **M1: Theory-practice gap.** We agree the algorithm departs from the idealized formulation, which we acknowledge in Section 3. Theorem 1 establishes the optimality condition (characterizing the Nash equilibrium), not a convergence guarantee. Empirically, the defender's harmlessness win rate stabilizes near 100% while attacker success declines (Figure 4(f,g)), consistent with NE-like convergence. The refusal reward $R_{D,refusal}$ is part of the zero-sum game structure, so blanket refusal incurs negative reward on benign queries, ruling out trivial equilibria. The reward shaping terms (format/revision adherence) are not zero-sum but serve as auxiliary signals that do not substantially alter the game dynamics. SFT co-training serves as regularization preserving general capability.
>
> ---
>
>
> **M4/Q2: Hyperparameters.** r_harm=r_refusal=1.0, r_format=1.0, r_revision=0.5, SFT coefficient=1.0. Full RL hyperparameters in Appendix A.2. The same hyperparameters transferred across all four base models without per-model tuning. We will add a consolidated table with sensitivity analysis.
>
> ---
>
>
> **M3/Q5: Multi-turn.** Thank you for raising this point. We agree that multi-turn is a promising direction we are excited to explore. Single-turn training is a deliberate scope decision: our contribution is the first online self-play MARL system for LLM safety, orthogonal to the single/multi-turn axis. Notably, our models already reduce multi-turn X-Teaming ASR: 88%→62% (3B), 76%→66% (14B), suggesting safety patterns transfer across formats. Tool-use prompt injection and system prompt extraction require fundamentally different threat models with multi-turn tool-calling infrastructure, which our framework does not assume. The self-play framework can naturally extend to multi-turn rollouts in future work and we are excited about this direction.
>
> ---
>
>
> **m1: Metric definitions.** Thank you for this suggestion. **ASR** = Attack Success Rate (↓ better); **RTA** = Robustness to Attacks, i.e., rate of safe responses (↑ better); **XSTest Compliance** = correctly answering benign-but-sensitive queries (↑ better, measures over-refusal); **AlpacaEval-2** = instruction-following capability (↑ better). Column suffixes: AH/VH = adversarial/vanilla harmful, AB/VB = adversarial/vanilla benign. We will add a consolidated table in the camera-ready.
>
> ---
>
>
> **m2: "First online MARL" claim.** Self-RedTeam is the first fully online, single-policy, full-parameter MARL framework for LLM safety. **Both AdvGame and WaltzRL cite our work**, confirming our temporal priority.
>
> ---
>
>
> **m3: Compute overhead.** Thank you for bringing this up. A notable aspect of our framework is the single shared-policy design: one model and one vLLM engine serve both attacker and defender roles, avoiding loading/offloading separate parameters. The 44-48% time overhead scales linearly with generated data (proportional to 50% more samples) with no additional GPU memory. This overhead yields substantial safety gains (Table 2: Self-Play+SFT outperforms Defender-Only+SFT by 17.33%).
>
> ---
>
>
> **E2/E4: Dual-use.** We commit to gated access with tracked usage and responsible use agreements. **E3:** We will add content warnings to appendix tables containing attack examples.

---

> > ### Author Rebuttal · Reviewer_XKFX · 2026-04-02
> >
> > Thank you for the detailed rebuttal. The clarifications on the reward hyperparameters, train/eval split, and the added cross-judge evidence address several of my original concerns.
> >
> > My remaining concern is mainly about scope: the theoretical guarantee applies to an idealized zero-sum game, while the implemented training objective and most static evaluations still rely on automated guard models. I would therefore appreciate a clearer scoping of the claims around the theorem and around judge-independent robustness.
> >
> > With that clarification, I would be open to revisiting my score.

---

> > > ### Author Response · Authors · 2026-04-07
> > >
> > > Thank you for your thoughtful engagement with our rebuttal and for clearly articulating the remaining concern. We are glad that the clarifications on reward hyperparameters, train/eval splits, and cross-judge evidence were helpful.
> > >
> > > ### 1. Scoping Theorem 1
> > >
> > > We appreciate this concern and agree that the manuscript would benefit from clearer scoping of the theorem relative to the implemented training procedure.
> > >
> > > **What Theorem 1 establishes.** Theorem 1 characterizes the *optimality condition* of the safety game: at Nash Equilibrium, the defender produces safe responses to any adversarial input. It motivates co-evolution, since optimizing only one side does not yield this guarantee.
> > >
> > > **Bridging theory and practice: a constrained zero-sum formulation.** As the reviewer correctly observes, our implemented training objective includes auxiliary terms (format adherence, revision faithfulness) alongside the core safety reward. Rather than treating these as modifications to the payoff that break the zero-sum structure, we can formally reinterpret them as *constraints on the policy space*.
> > >
> > > Concretely, define:
> > > - C_A = the set of valid attacker policies (e.g., must follow the prescribed reasoning format, must be capable of producing benign prompts), and
> > > - C_D = the set of valid defender policies (e.g., must follow the response format, must not refuse benign queries).
> > >
> > > The game then becomes a *constrained* two-player zero-sum game:
> > >
> > > $$\min_{\pi_A \in C_A} \max_{\pi_D \in C_D} \mathbb{E}_{y_A \sim \pi_A,\; y_D \sim \pi_D(\cdot \mid y_A)} \big[r\_\phi(y\_A, y\_D)\big]$$
> > >
> > > where r_φ is the core safety reward (harmfulness and refusal) only. Because the payoff remains purely zero-sum, the same NE safety guarantee from Theorem 1 applies: at equilibrium of this constrained game, the defender is safe against any attacker policy within C_A.
> > >
> > > In practice, hard constraints on neural network policies are infeasible, so our algorithm replaces them with soft penalty terms. Each player solves a regularized objective:
> > >
> > > - Attacker: $\min_{\pi_A} \; \mathbb{E}[r_\phi] + \lambda \cdot \text{Reg}(\pi_A)$
> > > - Defender: $\max_{\pi_D} \; \mathbb{E}[r_\phi] - \lambda \cdot \text{Reg}(\pi_D)$
> > >
> > > where $\text{Reg}$(π) ≥ 0 measures the degree to which π violates the constraints (e.g., format reward and revision faithfulness serve exactly this role), and Reg(π) = 0 when π satisfies all constraints.
> > >
> > > This formulation provides a clean formal bridge between the theory and practice:
> > > - When λ → ∞, the regularized game becomes exactly the constrained zero-sum game, and Theorem 1 applies directly.
> > > - When λ is finite but sufficiently large, the solution of the regularized game has bounded approximation error to the NE of the constrained game, a standard result in penalty-method optimization ([Bertsekas, 1999](https://books.google.com/books/about/Nonlinear_Programming.html?id=rC1EEAAAQBAJ), [Paternain et al., 2019](https://arxiv.org/pdf/1910.13393) ).
> > >
> > > In our implementation, r_harm and r_refusal constitute the zero-sum core r_φ, while r_format and r_revision serve as $\text{Reg}$ terms enforcing format and revision constraints. **The empirical evidence**, where the defender's harmlessness win rate stabilizing near 100% (Figure 4f) and consistent safety gains across 14 benchmarks, suggests our regularization strength is sufficient for convergence toward the safe-defender region predicted by the constrained game's equilibrium.
> > >
> > > We note that this constrained-game interpretation does not change the theorem or its validity — Theorem 1 holds as stated. The constrained formulation simply makes explicit how the auxiliary reward terms relate to the zero-sum structure that was always present in the algorithm. We will clarify Section 3 to make this connection explicit, so the scope of Theorem 1 as characterizing the optimality condition of the constrained game is clear to the reader.
> > >
> > > ### 2. Judge-Independent Robustness
> > >
> > > We are glad the cross-judge evidence was helpful. To further clarify scope: our paper includes multiple lines of judge-independent evidence. Most directly, we train with Qwen3Guard (independent from WildGuard) and evaluate with WildGuard — the model improves on 8 of 11 metrics (Table 3). If Self-RedTeam were merely learning to satisfy a specific judge, training with a different judge family should hurt WildGuard-evaluated performance; instead, we observe the opposite. Additionally, dynamic attacks using entirely independent judges (HarmBench-Llama-2-13b, GPT-4o) show consistent ASR reductions (Table 4), and benchmarks with their own predefined labels (XSTest, StrongREJECT, DAN, OR-Bench) confirm improvements independently of any guard model (Table 1). We will add a consolidated paragraph in the camera-ready cataloguing this evidence.
> > >
> > > ---
> > >
> > > We hope these address your remaining concerns. We are grateful for the constructive dialogue throughout this review process, and we would be glad if the reviewer considers these points in their final assessment.

---

### Official Review · Reviewer_YnRz · 2026-03-12

**Soundness:** 3
**Presentation:** 3
**Significance:** 2
**Originality:** 2
**Overall Recommendation:** 4
**Confidence:** 4

**Summary:**

This paper proposes an online self-play multi-agent reinforcement learning framework for safety alignment. The authors model red-teaming as a two-player zero-sum game between an attacker and a defender. They treat safety alignment as a dynamic co-evolutionary process rather than a static attack setting. The main contribution is SELF-REDTEAM, an online RL method in which a shared language model alternates between attacker and defender roles. The training objective is derived from a zero-sum formulation with a reward model that evaluates harmfulness and refusal behavior. The method is evaluated across multiple Llama and Qwen models and sizes. Results show increased diversity of discovered attacks compared to attacker-only training and improved robustness across different safety benchmarks.

**Compliance With Llm Reviewing Policy:**

Affirmed.

**Final Justification:**

The rebuttal addresses (1) limited baseline comparisons, (2) potential judge bias, and (3) the validity of the diversity claim by providing cross-judge evaluations, additional analysis on strategy-level diversity, and more detailed comparisons. However, some key empirical comparisons with strong training-time baselines remain incomplete. Overall, the rebuttal strengthens my confidence in the paper, but I keep my original assessment.

**Key Questions For Authors:**

Questions are mentioned in the weaknesses section.

**Limitations:**

Yes.

**Strengths And Weaknesses:**

Strengths
- In terms of originality, the paper reframes safety alignment as an online adversarial game. This problem formulation introduces new approach to the jailbreaking problem because it integrates attacker and defender training into a unified optimization objective. The theoretical usage of Nash equilibrium gives a principled interpretation of the training goal.
- In terms of significance, the paper shows improvements of safety alignment on jailbreak benchmarks on Qwen2.5 and Llama. The authors also conduct evaluation on benign benchmarks for over-refusal and instruction following benchmark, which strengthens the empirical claims.

Weaknesses
- While the empirical gains are strong, the comparative evaluation is limited. It does not present comparisons with strong existing defender methods [1,2,3] as well as similar methods that train defender models collaboratively [4].
- The evaluation relies on automated judge models (WildGuard-7B and Qwen3Guard), with no human evaluation or closed-source models. Since the same class of safety classifiers provides both the training reward and evaluation metrics, there is a risk of reward-model alignment rather than safety improvement.
- The diversity claim is supported primarily by semantic embedding (e.g., SBERT similarity). High semantic diversity does not necessarily imply meaningful strategic diversity. Increased embedding spread may reflect topic or intention drift rather than fundamentally new attack mechanisms.

[1]: AutoDefense: Multi-Agent LLM Defense against Jailbreak Attacks

[2]: X-Boundary: Establishing Exact Safety Boundary to Shield LLMs from Multi-Turn Jailbreaks without Compromising Usability

[3]: Steering Dialogue Dynamics for Robustness against Multi-turn Jailbreaking Attacks

[4]: Lifelong safety alignment for language models

---

> ### Author Rebuttal · Authors · 2026-03-31
>
> We thank Reviewer YnRz for recognizing the originality of our problem formulation and the significance of our empirical results. We appreciate the thoughtful suggestions and address each concern below.
>
> ---
>
> **W1: Limited comparative evaluation.**
>
> Thank you for suggesting these additional baselines.
>
> (a) Inference-time defenses (AutoDefense [1], NBF-LLM [3]): We agree these are valuable contributions. Because they are post-hoc output filters that do not modify model weights, they can be naturally layered on top of training-time methods like ours. A direct comparison would conflate two different defense layers, but we will discuss this complementarity in the camera-ready.
>
> (b) Training-time defenses (X-Boundary [2], Lifelong SA [4]):
> - We compare against CircuitBreaker (same representation-editing category as X-Boundary) in Appendix D.2 (Tables 13-14), and will move these to the main text. CircuitBreaker preserves capability but yields limited safety gains (WG:Test AH: 0.297 vs. base 0.303). We are also running X-Boundary and will provide results once available.
> - Lifelong SA is a very insightful suggestion, as it also co-trains attacker and defender. Unfortunately, due to GPU and time constraints, direct comparison is hardly feasible: its pipeline requires hosting R1-Qwen-32B, Qwen2.5-72B, and Llama-Guard-3-8B alongside the defender, and its checkpoints are LoRA adapters on Llama-3 (different from our models). We provide an analytical comparison:
>
> | | Self-RedTeam | Lifelong SA |
> |--|--|--|
> | Training | Online RL (REINFORCE++) | Offline SFT (iterative) |
> | Architecture | Single shared-policy | Separate attacker (32B) + defender (8B) |
> | Total parameters | 7-14B | 112B (32B+72B+8B) |
> | Co-adaptation | Continuous (simultaneous) | Discrete rounds |
>
> Self-RedTeam achieves strong results with 8-16x fewer parameters while enabling continuous co-adaptation. Appendix D.2 further shows the safety-utility tradeoff: LLM-LAT achieves near-perfect safety but catastrophic over-refusal (XSTest compliance drops to 0.004 on Llama and 0.552 on Qwen; AlpacaEval collapses to 9.68). CircuitBreaker avoids this but provides minimal safety improvement. Self-RedTeam maintains compliance (0.948, 0.964) while substantially improving safety across all metrics, achieving the balance that neither baseline manages. We will move these comparisons to the main text in the camera-ready.
>
> ---
>
> **W2: Judge circularity.**
>
> We want to directly clarify that our results are not limited to evaluating with the same judge used for training. We apologize that the manuscript does not make this sufficiently clear, and will revise it to address this misconception.
>
> Specifically: (1) In Table 3, we trained Qwen2.5-7B with Qwen3Guard (independent from WildGuard), then evaluated with WildGuard. The Qwen3Guard-trained model improves 8 of 11 metrics (avg +4.0%). If WildGuard were biased toward models trained on it, the opposite should hold. (2) Dynamic benchmarks (Table 4) use entirely independent judges: PAIR/AutoDAN use HarmBench-Llama-2-13b, X-Teaming uses GPT-4o. PAIR ASR: 0.218→0.108 (14B), AutoDAN: 0.555→0.336 (7B). (3) No train/eval overlap: training uses vanilla WildJailBreak partitions; evaluation uses structurally different adversarial partitions plus 5 independent benchmarks. We are also exploring human evaluation for the camera-ready, subject to IRB approval and annotator availability.
>
> ---
>
> **W3: Diversity claim.**
>
> To directly address your question, we conducted new experiments using GPT-4o strategy classification with WildTeaming's 35-tactic taxonomy (Jiang et al., 2024) at iterations 190-200:
>
> | | Self-Play | Attacker-Only |
> |--|--|--|
> | Strategy entropy (H) | 2.20 | 0.90 |
> | Concentration on dominant tactic | 29.8% (67/225) | 86.3% (214/248) |
> | Effective strategy count (2^H) | 4.6 | 1.9 |
>
> Attacker-Only collapses so that 86.3% of attacks use one dominant tactic (effective count: 1.9), while Self-Play maintains diversity with no tactic exceeding 30% (effective count: 4.6). Results hold across benign seeds (entropy 2.29 vs 1.71) and RL-Only self-play (2.08 vs 0.90). Base-model perplexity on Self-Play attacks is 24.74 vs. 8.62 on base-model attacks (Table 15), nearly 3x higher, providing complementary evidence of substantive novelty beyond surface paraphrases. We will include this strategy classification analysis and qualitative attack examples in the camera-ready.
>
> We hope these responses address the reviewer's concerns. We are grateful for the constructive feedback and will incorporate all suggested improvements in the camera-ready version. We would be glad if the reviewer considers these points in their final assessment.

---

> > ### Author Rebuttal · Reviewer_YnRz · 2026-04-03
> >
> > The rebuttal addresses my concerns well. In particular, the authors provide additional judge evaluations to mitigate the circularity issue and introduce a more meaningful strategy-level analysis to support the diversity claim. I'd like to keep my original assessment.

---

> > > ### Author Response · Authors · 2026-04-07
> > >
> > > We sincerely thank the reviewer for the thoughtful engagement with our rebuttal. We're glad the additional judge evaluations and strategy-level diversity analysis were convincing.
> > >
> > > We noticed the reviewer selected option (a), which includes *"please consider adjusting your score accordingly."* Since the core concerns around circularity and diversity have been fully resolved, we would really appreciate it if the reviewer could raise their score to reflect this. We believe this paper makes a meaningful contribution and your support would go a long way in advocating for it.
> > >
> > > Thank you again for the constructive feedback, it genuinely helped strengthen the work.

---

### Official Review · Reviewer_ALh2 · 2026-03-13

**Soundness:** 3
**Presentation:** 4
**Significance:** 3
**Originality:** 3
**Overall Recommendation:** 4
**Confidence:** 4

**Summary:**

Traditional LLM safety alignment relies on a disjoint, reactive loop where attackers exploit static models, and defenders patch vulnerabilities post-hoc. This sequential approach leads to attackers perpetually staying ahead, exploiting newly discovered threats before defenses can adapt, resulting in incomplete vulnerability coverage and safety degradation. The paper proposes self-redteam, the first fully online self-play multi-agent reinforcement learning algorithm for LLM safety. It employs a single model to co-evolve as both attacker and defender, enabling continuous adaptation to emerging threats. The authors also make theoretical connections showing that in theory they can reach safe LLM, grounding the algorithm's justification.

**Compliance With Llm Reviewing Policy:**

Affirmed.

**Final Justification:**

I acknowledge and thank the authors for the rebuttal. Given the responses and feedback from the rest of the reviewers, I feel the current score is justified. Therefore, I will maintain my score.

**Key Questions For Authors:**

N/A

**Limitations:**

Yes

**Strengths And Weaknesses:**

I see the following strengths and weaknesses in the paper.

Strengths:

1/ Novelty of the idea: I find the idea to be quite novel and addressing a critical gap in today's training. Infusing attack optimization during training makes a lot of sense to provide better guardrailed models.

2/ Theoretical Safety Guarantees: The paper provides a theoretical foundation by framing LLM safety alignment as a zero-sum game and proving that convergence to a Nash Equilibrium ensures safe responses against any adversarial input. While the authors call out that practically the NE may not be achievable, this part of the contribution is appealing as it grounds the proposed algorithm.

3/ Rigorous experimentation showing strong results: The authors show results on a variety of datasets and models. While some models are old now, I think there is enough evidence that the method works well.

Weaknesses:

1/ It makes sense that one of the motivations for the paper is that human red-teamers supplying prompts to train models is slow. However, I do believe that automated methods can not completely plug the novelty that some human red team designed prompts bring. In particular, red teamers are good at finding gaps in the model at hand and providing valuable prompts that can be used in patching models in next round of training. It would had been good to discuss how human provided prompts can be mixed in an algorithm like this.

2/ The method could be quite expensive as it requires multiple optimizations. It would be good to discuss how this optimization could be mixed in with regular utility training (e.g. GRPO on STEM, agents etc.) and what the training cost increase would be compared to a baseline where vanilla safety training is done using a given set of prompts.

---

> ### Author Rebuttal · Authors · 2026-03-31
>
> We thank the reviewer for the positive evaluation and for recognizing the novelty of our formulation, theoretical grounding, and rigor of our experimental evaluation. We address the raised concerns below.
>
> ---
>
> **W1: Integration with Human Red-Teamers**
>
> We fully agree that skilled human red-teamers bring unique value. They can uncover especially novel, model-specific failure cases that automated methods may miss. We view human- and automatically generated red-teaming prompts as complementary rather than competing, and appreciate the reviewer highlighting this important practical consideration.
>
> Human-written attacks can be naturally incorporated into Self-RedTeam. As shown in Algorithm 1, the training loop samples from a seed prompt dataset $\mathcal{D}_{RL}$ that is **source-agnostic**. It accepts any initial prompts for the attacker to revise. The choice to use 26,000 prompts from WildJailBreak was a design decision for our experiments, not an algorithmic constraint. Human-crafted adversarial prompts can be used to initialize or replace these seeds, and the attacker then builds on them to generate further variations through self-play. Beyond one-time seed injection, human red-teamers can also be deployed periodically to identify new failure modes, which are then fed back as seeds for continued self-play training. In this sense, Self-RedTeam is designed to **amplify the value of scarce human expertise** by using it as a starting point for scalable adversarial exploration. We will expand on this discussion in the camera-ready.
>
> ---
>
> **W2: Computational Cost and Utility Co-Training**
>
> Thank you for raising this important practical consideration. The 44-48% time overhead of the self-play-only variant (Appendix D.3) scales **linearly** with the amount of generated data — proportional to the 50% growth in training samples — representing near-optimal scaling. A key advantage of Self-RedTeam's single shared-policy design is that it avoids sequential loading/offloading between separate attacker and defender parameters: one model and one vLLM engine serve both roles, eliminating the coordination overhead that multi-model adversarial frameworks (e.g., Lifelong SA with 32B attacker + 8B defender) require. There is no additional GPU memory overhead beyond standard online RL.
>
> **Utility co-training**: Self-RedTeam already co-trains RL and SFT with joint gradient accumulation (Algorithm 1; Appendix A.2), using 30K SFT examples (15K WildJailBreak benign + 15K HelpSteer3). Table 1 confirms capability preservation: AlpacaEval-2 improves for Qwen models (+0.3% to +4.5%). The framework is compatible with different SFT datasets (e.g., STEM/reasoning) and alternative reward models (Table 3 demonstrates seamless judge-swapping with Qwen3Guard); we leave systematic exploration of these extensions to future work. Details are in Appendix A.2 and D.3.

---

> > ### Author Rebuttal · Reviewer_ALh2 · 2026-04-03
> >
> > I acknowledge and thank the authors for the rebuttal. Given the responses and feedback from the rest of the reviewers, I feel the current score is justified. Therefore, I will maintain my score.

---

> > > ### Author Response · Authors · 2026-04-07
> > >
> > > We thank the reviewer for acknowledging the rebuttal. We're glad our responses on human red-teamer integration and computational cost were satisfactory, both were thoughtful questions that improved our discussion.
> > >
> > > We noticed the reviewer selected "follow-up questions" but didn't raise any specific ones. We'd be happy to address any remaining points if there are questions we missed.
> > >
> > > Regarding the other reviewers' feedback: we want to highlight that the key cross-reviewer concerns have been substantively addressed in our rebuttals. Specifically:
> > >
> > > 1. Judge circularity was mitigated with three independent lines of evidence, including Qwen3Guard cross-judge evaluation improving 8 of 11 metrics and dynamic benchmarks using entirely separate judges (HarmBench-Llama-2-13b, GPT-4o);
> > >
> > > 2. Strategic diversity was validated with a new GPT-4o strategy classification showing 2.4x higher effective strategy count for Self-Play vs. Attacker-Only; and
> > >
> > > 3. Safeguarding baselines (LLM-LAT, CircuitBreaker) are already compared in Appendix D.2 and will be moved to the main text.
> > >
> > > We hope this additional context is reassuring, and we would welcome any specific follow-up questions the reviewer may have.

---

### Official Review · Reviewer_khNb · 2026-03-13

**Soundness:** 3
**Presentation:** 2
**Significance:** 3
**Originality:** 3
**Overall Recommendation:** 5
**Confidence:** 4

**Summary:**

Authors propose SELF-REDTEAM as a method that uses online self-play MARL within a single language model to co-train an attacker (red-team) role and a defender role within the same LLM. The attacker generates adversarial prompts such as hidden chain-of-thought strategies, while the defender learns to refuse unsafe requests; in self-play fashion, both roles share a single set of weights updated via Re++ with role-conditioned system prompts. The method is evaluated across a few model configurations on 14 safety benchmarks and two capability benchmarks. A theoretical result frames the setup as a two-player zero-sum game and claims a safety guarantee at Nash equilibrium.

**Compliance With Llm Reviewing Policy:**

Affirmed.

**Final Justification:**

The authors did a thorough job at addressing my questions and doubts regarding various different concerns I had. I find their contribution worthy of acceptance.

**Key Questions For Authors:**

1. What are the magnitudes of $r_{\text{harm}}$, $r_{\text{refusal}}$, $r_{\text{format}}$, $r_{\text{revision}}$, and the SFT weighting $\alpha$?

2. Can you evaluate with a safety classifier NOT used during training (see weaknesses)?

3. Have you noticed any cycling in the strategies? Would you expect it under further optimization pressure?

**Limitations:**

(1) WildGuard circularity; (2) underdiscussed capability loss (Llama shows 13.5% AlpacaEval degradation vs. Qwen's improvement); (3) the gap between the theoretical framing (zero-sum game, NE convergence) and the actual training procedure (non-really-zero-sum rewards, Re++ without convergence guarantees).

**Strengths And Weaknesses:**

**Strenghts**

1. The strongest part of the paper is the (empirical) observation that co-evolution of attackers and defenders via selfplay beats static-data defender training on the authors' own benchmark suite. Table 1 shows harmfulness reductions across the Qwen family and meaningful gains on the safety-ablated Llama model. Table 4 also gives some evidence that the method transfers beyond the exact training distribution. The benign-seed design is neat, sensible and well motivated.

2. Co-evolution of attackers and defenders is a plausible direction for safety training (generally in data-scarce regimes too, if done carefully), and a single-policy formulation that does not require training two separate large models is of pragmatic relevance too. The paper also does more than just propose a toy mechanism, and lays down the engineering, which is laudable (modulo some things, see below).

3. The 6-way ablation is the paper's strongest empirical contribution, because it clearly isolates the effect of self-play vs. attacker-only vs. defender-only training, and separately tests hidden CoT, revision faithfulness, and benign seeding. Figure 4's training dynamics shows informative patterns: the attacker's ASR rises then falls as the defender adapts, and diversity increases monotonically under self-play but plateaus under attacker-only training.

---

**Weaknesses**

1. The static safety evaluation is documented in a very dubious way. The model is trained directly against automated harmfulness/refusal labels from a safety judge (WildGuard-7b), and the paper then appears to score benchmark outputs using another automated harmfulness/refusal classifier, but never specifies whether this evaluator is independent/different. Because the paper uses the Ai2 Safety Tool / safety-eval stack for safety evaluation, and that stack defaults to WildGuard for many safety benchmarks unless otherwise specified. If the same judge, or the same judge family, is used on both sides, then the method can look good by learning how to satisfy that classifier rather than by becoming robust in any sense. **Right now the paper does not rule out judge-specific reward hacking.** (or at the minimum, it is not clear to me this is the case).

2. +17.80% SBERT diversity claim is highlighted but not precisely reported. My guess is that the authors have reverse-engineered the number from Figure 4(e) by reading off convergence values (Self-Play + SFT ≈ 0.855, Attacker-Only ≈ 0.725, relative difference ≈ 17.9%), but this means an abstract headline number is derived from eyeballing a training curve at a particular step (with Attacker-Only having large confidence intervals as well). Table 15 -- the only table reporting SBERT -- omits Attacker-Only entirely and shows only +3.7-4.1% improvement over the base model. The abstract's claim reads as "over baseline", not over an ablation, and is rather counterintuitive. This number should likely be tabulated.

3. Main-text comparisons (Tables 1-3) only pit SELF-REDTEAM against the undefended base model. LLM-LAT and CircuitBreaker appear only in Appendix Table 13, tested on 2 of 5 model configurations. This makes it impossible to understand whether the gains come from the self-play mechanism specifically or from any additional safety training. The method may be novel, but the ~apples-to-apples comparisons are delegated to the appendix only.

4. Self-BLEU and SBERT capture surface diversity, as opposed to "strategy" (in NL sense) diversity. What I mean by strategy is whether attacks exploit different jailbreak strategies (e.g., role-play, encoding, few-shot manipulation, etc.). The self-play examples in Figure 3 differ in topics (nuclear weapons, slurs, mental health) but technique-wise they're quite narrow: most are direct requests with framing variations. I thus find the paper's claim of "genuinely novel strategies" (§6) to be supported only by surface metrics.

---

> ### Author Rebuttal · Authors · 2026-03-31
>
> We thank Reviewer khNb for the detailed and constructive feedback, and for recognizing the comprehensiveness of our experimental setup. We address each point below.
>
> ---
>
> **W3 (Safeguard baselines):** We appreciate this suggestion. Tables 13-14 in the appendix already compare against LLM-LAT and CircuitBreaker on Llama-3.1-8B and Qwen2.5-7B, and **we will move these to the main text** in the camera-ready. Key finding: LLM-LAT achieves near-perfect safety but catastrophic over-refusal (XSTest compliance: 0.924→0.004 Llama, 0.948→0.552 Qwen); CircuitBreaker preserves capability but yields limited safety gains (WG:Test AH: 0.297 vs. base 0.303). Self-RedTeam maintains compliance (0.948, 0.964) while substantially improving safety. Table 2 further shows Self-Play+SFT achieves 17.33% additional safety improvement over Defender-Only+SFT, demonstrating the value of co-evolution beyond simply training on more safety data.
>
> ---
>
> **W4 (Surface vs. strategic diversity):** To directly address your question, we conducted new experiments using GPT-4o strategy classification with WildTeaming's taxonomy (Jiang et al., 2024) at iterations 190-200:
>
> | | Self-Play | Attacker-Only |
> |--|--|--|
> | Strategy entropy (H) | 2.20 | 0.90 |
> | Concentration on dominant tactic | 29.8% (67/225) | 86.3% (214/248) |
> | Effective strategy count (2^H) | 4.6 | 1.9 |
>
> Attacker-Only collapses so that 86.3% of attacks use one dominant tactic (effective count: 1.9), while Self-Play maintains diversity with no tactic exceeding 30% (effective count: 4.6). Results hold on benign seeds (entropy 2.29 vs 1.71) and RL-Only self-play (2.08 vs 0.90), confirming co-evolution drives this effect. Base-model perplexity on Self-Play attacks is 24.74 vs. 8.62 on base-model attacks (Table 15), providing complementary evidence of substantive novelty in attack generation. We will include this strategy clustering analysis in the camera-ready.
>
> ---
>
> **W1/Q2 (Judge circularity):** We want to clarify that our results are not limited to evaluating with the same judge used for training. We apologize for the clarity issues that led to this misconception, and will revise the manuscript. Three lines of evidence: (1) Table 3 trains with Qwen3Guard (independent from WildGuard), then evaluates with WildGuard. The Qwen3Guard-trained model improves 8 of 11 metrics. If WildGuard were biased toward models trained on it, the opposite should hold. (2) Dynamic benchmarks (Table 4) use entirely independent judges: PAIR/AutoDAN use HarmBench-Llama-2-13b; X-Teaming uses GPT-4o. PAIR ASR: 0.218→0.108 (14B), AutoDAN: 0.555→0.336 (7B). (3) No train/eval overlap: training uses vanilla WildJailBreak partitions; evaluation uses structurally different adversarial partitions plus 5 independent benchmarks.
>
> ---
>
> **W2 (SBERT +17.80%):** Thank you for raising this question. You are correct: the +17.80% is the relative difference between Self-Play (~ 0.855) and Attacker-Only (~ 0.74) convergence values in Figure 4(e) during training. Table 15 shows evaluation-time (on a holdout set of 100 prompts) SBERT diversity improvement is +3.7-4.1% over the base model. We will clarify this in the camera-ready.
>
> ---
>
> **Q1 (Reward magnitudes and SFT alpha):** Zero-sum: r_harm=r_refusal=1.0. Shaping: r_format=1.0, r_revision=0.5. SFT coefficient=1.0. Full RL hyperparameters in Appendix A.2. The same hyperparameters transferred across all four base models without per-model tuning. We will add a consolidated table with sensitivity analysis.
>
> ---
>
> **Capability loss:** The Llama-3.1-8B AlpacaEval decrease (-13.5%) is model-specific. All Qwen models improve: +0.3% (3B), +4.5% (7B), +3.2% (14B). General benchmarks (ARC-C, GPQA, MMLU, TruthfulQA) show minimal degradation (Appendix C). Llama-specific degradation is consistent with this family's known sensitivity to safety fine-tuning, where stronger safety tends to increase over-refusal (Cui et al., 2024; Bianchi et al., 2024).
>
> ---
>
> **Theory-practice gap:** We agree the algorithm departs from the idealized formulation. Theorem 1 establishes the optimality condition motivating our objective, not a convergence guarantee. Though empirically the defender's harmlessness winrate stabilizes near 100% (Figure 4(f,g)), consistent with NE-like convergence. The refusal reward is part of the zero-sum structure, so blanket refusal incurs negative reward on benign queries. The reward shaping terms (format/revision) are not zero-sum but serve as auxiliary signals that do not substantially alter the game dynamics.
>
> ---
>
> **Q3 (Cycling between attacker/defender strategies):** Figure 4 shows near monotonic improvement with no observed cycling. Even if cycling occurred, the defender outperforms baselines at each checkpoint throughout training. We acknowledge investigating cycling under extended optimization as interesting future work.

---

> > ### Author Rebuttal · Reviewer_khNb · 2026-04-03
> >
> > I appreciate the authors for their thorough response.
> >
> > I have one question: what do you mean by "we conducted new experiments using GPT-4o strategy classification with WildTeaming's taxonomy"? Could you explain how you conducted this evaluation?

---

> > > ### Author Response · Authors · 2026-04-07
> > >
> > > Thank you for the follow-up and are happy to clarify the strategy classification methodology. We will add the following clarification to the appendix of the camera-ready paper.
> > >
> > > **Methodology.** We adapted the GPT-4o-based strategy classification pipeline from WildTeaming ([Jiang et al., 2024](https://arxiv.org/pdf/2406.18510)), and WildTeaming is a peer-reviewed work published in NeurIPS’24 with plenty of followup work. This pipeline provides a published taxonomy of 35 jailbreak strategies with formal definitions (e.g., "contextualizing the task," "roleplay as an evil bot," "implied harm," "providing seed examples"; see Table 7-8 in [Jiang et al., 2024](https://arxiv.org/pdf/2406.18510)). The procedure is as follows:
> > >
> > > 1. **Sampling.** We sampled ~250 generated attacks from Self-Play and Attacker-Only checkpoints at iterations 190–200 using **the same random seed and the same set of seed prompts**. These generate attacks are a mix of harmful and benign revision objectives (the attacker must revise both types of seeds).
> > >
> > > 2. **Classification.** For each generated attack, we provided GPT-4o with: (a) the original seed prompt (the "simple harmful prompt" in WildTeaming's terminology), (b) the model-generated adversarial revision, and (c) the full list of 35 WildTeaming strategies with their definitions. Following WildTeaming's prompt structure, GPT-4o analyzes how the seed was revised into the adversarial prompt and matches the revision to strategies from the taxonomy. We adapted the classification to identify the single most dominant strategy per attack, yielding a clean distribution over strategy types.
> > >
> > > 3. **Aggregation.** We computed three summary statistics over the distribution of dominant strategies across all attacks for each method: (a) Shannon entropy H, measuring how evenly attacks spread across strategies; (b) concentration on the single most frequent tactic; and (c) effective strategy count (2^H), indicating how many strategies are meaningfully active.
> > >
> > > **Results.** The results **reveal a stark contrast in strategy distributions**:
> > >
> > > - **Attacker-Only** collapses to a single dominant tactic: 86.3% of attacks (214/248) are classified as "contextualizing the task" (i.e., embedding the harmful request in a detailed narrative context). The remaining attacks scatter thinly across "implied harm" (4.0%), "pseudonym/coded language" (3.6%), and others.
> > >
> > > - **Self-RedTeam** maintains a substantially more balanced distribution across multiple strategies: "downplaying the request with more nuanced expressions" (29.8%), "contextualizing the task" (27.6%), "implied harm" (24.4%), "elevating the moral grounding of a harmful request" (10.7%), and "pseudonym/coded language" (6.7%). No single tactic exceeds 30%.
> > >
> > > This pattern is consistent across benign seeds (strategy entropy 2.29 vs 1.71) and RL-Only self-play (strategy 2.08 vs 0.90), confirming that co-evolution specifically drives the diversity effect.
> > >
> > > We appreciate the reviewer's suggestion, as this strategy-level analysis nicely strengthens and contextualizes the diversity collapse already established in our submission via SBERT and Self-BLEU metrics. Together with the base-model perplexity gap (24.74 vs. 8.62 for Self-Play vs. Attacker-Only attacks, Table 15), indicating that Self-Play attacks are nearly 3x more surprising to the base model, these results confirm that self-play uncovers genuinely novel attack strategies, not merely surface paraphrases.
> > >
> > > We will include full methodology details and qualitative examples in the appendix of the camera-ready. We hope this clarification addresses the reviewer's question, and we would be grateful if the reviewer considers these points in their final assessment.

---

### Decision · Program_Chairs · 2026-04-30

**Decision:**

Accept (regular)

**Comment:**

This paper introduces Self-RedTeam, an online self-play MARL framework where a single LLM co-evolves as both attacker and defender for safety alignment, with hidden chain-of-thought reasoning and a zero-sum game formulation. All four reviewers were positive (one Accept, three Weak Accept), praising the novelty of the formulation, comprehensive evaluation across five models and 14 benchmarks, and the informative ablation study. Key concerns were substantively addressed in rebuttal: judge circularity was mitigated through cross-judge training (Qwen3Guard trained, WildGuard evaluated, improving 8 of 11 metrics) and dynamic benchmarks with independent judges; strategic diversity was validated via GPT-4o classification showing 2.4x higher effective strategy count for self-play; and the theory-practice gap was clarified by reinterpreting auxiliary rewards as soft constraints on the zero-sum core. Reviewer khNb raised their score to Accept, and all reviewers maintained positive assessments. Reviewer XKFX's ethics flag regarding prompt injection text is an ICML-placed watermark per official guidance.

Residual limitations are reasonable: the method is currently single-turn only, Llama models show some capability degradation versus Qwen's improvements, and comparisons with some baselines remain analytical due to compute constraints. The authors commit to moving existing baseline comparisons (LLM-LAT, CircuitBreaker) from the appendix to the main text.

Based on the positive consensus, thorough rebuttals, and the practical significance of the contribution, I recommend this paper for acceptance.

---

Note: the paper contains a reference with a wrong author list which is worth fixing.

```
Reference: Rein, D., Li, B., Lee, J. H., Steinhardt, J., and Hendrycks, D. GPQA: A graduate-level google-proof q&a benchmark. arXiv preprint arXiv:2311.12022, 2023. URL https://arxiv.org/abs/2311.12022.
```